# ViMoE: An Empirical Study of Designing Vision Mixture-of-Experts

## Abstract

Mixture-of-Experts (MoE) models embody the divide-and-conquer concept and are a promising approach for increasing model capacity, demonstrating excellent scalability across multiple domains. In this paper, we integrate the MoE structure into the classic Vision Transformer (ViT), naming it ViMoE, and explore the potential of applying MoE to vision through a comprehensive study on image classification. However, we observe that the performance is sensitive to the configuration of MoE layers, making it challenging to obtain optimal results without careful design. The underlying cause is that inappropriate MoE layers lead to unreliable routing and hinder experts from effectively acquiring helpful knowledge. To address this, we introduce a shared expert to learn and capture common information, serving as an effective way to construct stable ViMoE. Furthermore, we demonstrate how to analyze expert routing behavior, revealing which MoE layers are capable of specializing in handling specific information and which are not. This provides guidance for retaining the critical layers while removing redundancies, thereby advancing ViMoE to be more efficient without sacrificing accuracy. We aspire for this work to offer new insights into the design of vision MoE models and provide valuable empirical guidance for future research.

## 1 Introduction

General artificial intelligence is continuously developing toward larger and stronger models (Achiam et al., 2023; Yang et al., 2024; AI@Meta, 2024). However, larger models require significant computational resources for training and deployment, and balancing performance with efficiency remains a critical issue, especially in resource-constrained environments. A promising approach is to use the Mixture-of-Experts (MoE) (Jacobs et al., 1991) layers in neural networks, which decouple model size from inference efficiency. MoE embodies the *divide-and-conquer* principle, where feature embeddings are routed to selected experts through a gating mechanism, allowing each expert to specialize in a subsets of the data. As a result, each input is processed by only a small portion of the parameters, whereas traditional dense models activate all parameters for every input. This approach is becoming increasingly popular in Natural Language Processing (NLP), as it enables parameter scaling while keeping computational costs at a modest level (Jiang et al., 2024; Dai et al., 2024).

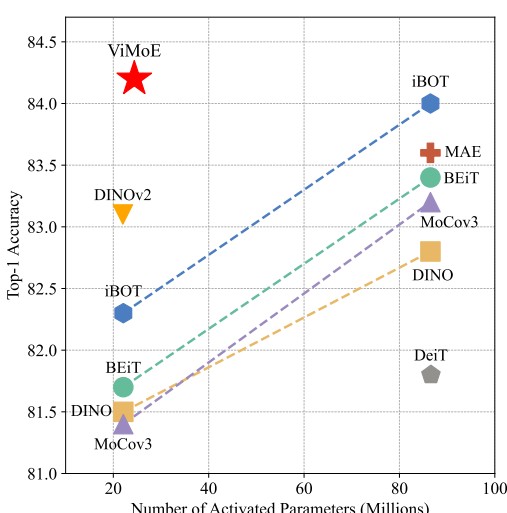

Figure 1: **Top-1 accuracy on ImageNet-1K.** We compare ViMoE with other ViT architecture baselines. All models are evaluated at $224 \times 224$.

This work focuses on exploring the simple application of MoE in vision models. We convert the classic Vision Transformer (ViT) (Dosovitskiy, 2020) into a sparse MoE structure, naming it Vi-

MoE. Our modification of ViT follows (Riquelme et al., 2021), where the Feed-Forward Networks (FFNs) in each block is replaced with multiple experts, while keeping the structure of each expert the same. For simplicity and efficiency, we choose to select experts at the image level rather than the token level (Daxberger et al., 2023; Liu et al., 2024). Through a comprehensive study on image classification, we explore strategies for configuring MoE in a stable and efficient manner, while also observing several interesting phenomena related to expert routing from different perspectives.

An essential consideration in designing ViMoE is determining how many MoE layers to include and where to position them. A common approach is to insert them into the last $L$ ViT blocks (Wu et al., 2022; Liu et al., 2024), which receive the largest gradient magnitudes. Alternatively, one more straightforward approach would be to add MoE layers to all blocks without careful design. We adopt an exhaustive way of scanning the number of layers to determine which configuration yields the optimal accuracy for ViMoE. Interestingly, increasing the number of MoE layers does not always lead to better performance; instead, a downward trend emerges beyond a certain number of layers. We attribute this to the fact that inappropriate MoE layers, particularly in the shallow ViT blocks, not only fail to contribute but also complicate optimization. While scanning and observing can reveal the optimal performance point and the most suitable number of MoE layers, such an approach is invariably laborious. Inspired by (Xue et al., 2022; Dai et al., 2024), we introduce a shared expert that absorbs knowledge from the entire dataset, alleviating the inadequacies in individual expert learning and the burden on the routing mechanism. The shared expert brings more excellent stability to ViMoE, as it prevents the accuracy degradation observed with an excessive number of MoE layers. This eliminates the need for constant trial and error to find the optimal point, thereby facilitating a more streamlined design process.

The above are deductions drawn from the scanning results, but we seek further heuristic exploration. Building on the stable ViMoE, we attempt to delve deeper into the routing behavior within MoE layers to uncover what each expert focuses on. Owing to our routing strategy, we can observe how data from each class are distributed across the experts. For the MoE layers in the deeper ViT blocks, the gating network effectively allocates samples of the same class to the same expert, with each expert specializing in processing different data. However, in the shallow blocks, the gating network struggles to consistently route images of the same class to the same expert or effectively guide the experts to specialize in different classes. This suggests that the experts have not learned highly discriminative knowledge; rather, they end up implementing very similar functions, indiscriminately extracting common features across all classes (Riquelme et al., 2021). These results highlights which layers truly fulfill the *divide-and-conquer* role and which do not, corresponding to the accuracy trends observed through layer scanning.

Furthermore, we aim to inform more thoughtful and efficient ViMoE designs through our observations of MoE behavior. One attempt we propose is to estimate the necessary number of MoE layers based on the routing distribution, and then combine this with the number of experts set per layer to approximate the required expert combinations. This insight allows us to simplify the structure by removing potentially redundant MoE layers, thereby achieving a more efficient ViMoE. As a result, our ViMoE based on ViT-S/14 outperforms DINOv2 (Oquab et al., 2023) by 1.1% on ImageNet-1K (Deng et al., 2009) fine-tuning. With less than one-third of the activated parameters, ViMoE even surpasses a number of advanced ViT-B/16 models (Bao et al., 2021; Touvron et al., 2021; Zhou et al., 2021; Zhang et al., 2022; Xinlei et al., 2021; He et al., 2022).

In summary, we believe that as MoE becomes more widely adopted in vision tasks, the observations, evidence, and analyses presented in this study are worth knowing. We hope that our insights and experiences will contribute to advancing this frontier.

## 2 VIMOE

### 2.1 PRELIMINARY

**Mixture-of-Experts (MoE)** (Jacobs et al., 1991; Jordan & Jacobs, 1994) is a promising approach that allows for scaling the number of parameters without increasing computational overhead. For Transformer-based MoE models, the architecture mainly consists of two key components: *(1) Sparse MoE Layer:* A MoE layer contains $N$ experts (denoted as $E_i(\cdot), i = 1, 2, \ldots, N$), each functioning as an independent neural network (Shazeer et al., 2017). *(2) Gating Network:* This component is

responsible for routing the input token $\boldsymbol{x}$ to the most appropriate top-$k$ experts (Cao et al., 2023). The gate consists of a learnable linear layer, defined as $g(\boldsymbol{x}) = \sigma(\boldsymbol{W}\boldsymbol{x})$, where $\boldsymbol{W}$ is the gate parameter, and $\sigma$ is the softmax function. Let $\mathcal{T}$ represent the set of the top-$k$ indices, and output of the layer is then computed as a linear combination of the outputs from the selected experts weighted by the corresponding gate values,

$$\boldsymbol{y} = \sum_{i \in \mathcal{T}} g_i(\boldsymbol{x}) \cdot E_i(\boldsymbol{x}). \tag{1}$$

**Load Balancing Loss.** To encourage load balancing among the experts, we incorporate a differentiable load balancing loss (Lepikhin et al., 2020; Zoph et al., 2022) into each MoE layer, promoting a more balanced distribution of input tokens across the experts. For a batch $\mathcal{B}$ containing $T$ tokens, the auxiliary loss is calculated as a scaled dot product between the vectors $f$ and $P$,

$$\mathcal{L}_{\text{aux}} = \alpha \cdot N \cdot \sum_{i=1}^{N} f_i \cdot P_i, \tag{2}$$

where $\alpha$ is the loss coefficient, $f_i$ represents the fraction of tokens routed to expert $i$, and $P_i$ is the fraction of the router probability assigned to expert $i$,

$$f_i = \frac{1}{T} \sum_{\boldsymbol{x} \in \mathcal{B}} \mathbf{1}\{\arg\max g(\boldsymbol{x}) = i\}, \tag{3}$$

$$P_i = \frac{1}{T} \sum_{\boldsymbol{x} \in \mathcal{B}} g_i(\boldsymbol{x}). \tag{4}$$

**MoE Transformer.** A widely used approach to applying MoE to Transformer models is to replace the Feed-Forward Networks (FFNs) in some of the standard (non-MoE) Transformer blocks with MoE layers (Fedus et al., 2022). Specifically, in an MoE layer, the experts retain the same structure as the original FFNs. The gating function receives the output from the preceding self-attention layer and routes the token representations to different experts.

## 2.2 Settings for Image Classification

**Architecture.** We introduce a ViMoE framework to facilitate our study on the application of MoE for image classification. We choose the Vision Transformer (ViT) (Dosovitskiy, 2020) backbone and replace the FFNs in the ViT blocks with MoE layers. Instead of training from scratch (Riquelme et al., 2021), we consider inheriting self-supervised pre-training weights, which reduces training costs while also benefiting from advanced feature representations. Since the experts in the MoE layers share the same structure as the FFNs, we simply replicate the pre-trained weights of the FFNs across each expert for initialization.

**Routing Strategy.** Recent large-scale sparse MoE models (Achiam et al., 2023; Jiang et al., 2024; Dai et al., 2024; Yang et al., 2024) typically employ a token-based routing strategy, where the gating mechanism assigns each token to selected experts. However, it is worth considering whether this strategy is necessary for MoE in image classification, where the model focuses more on the overall features of the image to predict a single class for the image. We suggest that the routing strategy should align with the specific requirements of the vision task. Routing at the image level (*i.e.*, selecting experts for each entire image) (Daxberger et al., 2023; Liu et al., 2024) is simpler and better suited to the objectives of image classification. In practice, we use the `[CLS]` token to represent the image $\boldsymbol{x}$ as the input to the gating network, since it encapsulates the information from all image tokens and is used for classification prediction. Additionally, unless otherwise specified, we default to selecting only the top-1 routed expert to simplify the architecture. Therefore, compared to token-based routing, this strategy reduces the number of experts activated per image.

**Shared Expert.** There is often some common sense or shared information across input tokens assigned to different experts. As a result, with a conventional routing strategy, multiple experts may acquire overlapping knowledge within their respective parameters. By designing the shared

expert (Xue et al., 2022; Dai et al., 2024) to focus on capturing and consolidating common information, other routed experts can specialize in learning unique knowledge, leading to a more parameter-efficient model composed of a greater number of specialized experts. Consequently, we introduce the shared expert $E_s(\cdot)$ into ViMoE to enable learning common knowledge from all data. In our implementation, we set the number of shared experts to 1, with a structure identical to that of the other experts. The output of the shared expert is added to the output of the selected routed expert, allowing Eq. 1 to be rewritten as,

$$y = E_s(\boldsymbol{x}) + \sum_{i \in \mathcal{T}} g_i(\boldsymbol{x}_{\texttt{[CLS]}}) \cdot E_i(\boldsymbol{x}). \tag{5}$$

## 3 EMPIRICAL OBSERVATIONS IN DESIGNING VIMOE

### 3.1 A STABILITY STRATEGY FOR CONVENIENT DESIGN

**Scanning the Number of MoE Layers.**
When designing ViMoE, an important consideration is determining how many MoE layers to include and where to place them within the ViT blocks. Here, we start by exploring sparse MoE without the shared expert for simplicity. The most straightforward approach is to place the MoE layer in every ViT block or to select the **last** $L$ blocks where the gradient magnitudes are the largest. To explore reasonable configurations and seek guiding insights, we scan the number of MoE layers and evaluate the accuracy of image classification. Our experiments are based on the DINOv2 (Oquab et al., 2023) pre-trained ViT-S/14 (Dosovitskiy, 2020), modified into ViMoE and fine-tuned on ImageNet-1K (Deng et al., 2009) for 200 epochs (more implementation details are provided in Sec. 4.1). From Fig. 2, it can be observed that regardless of the number of experts, whether $N = 2$, $N = 4$, or $N = 8$, the

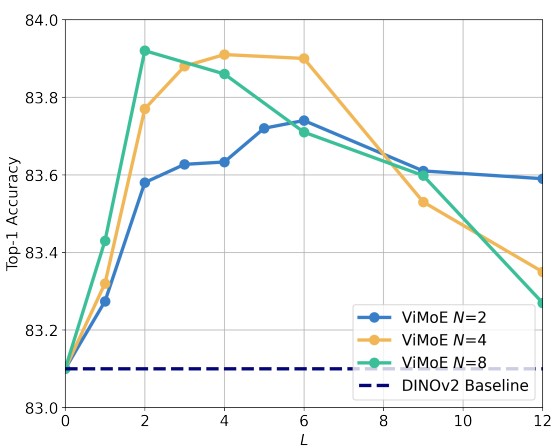

Figure 2: **Top-1 accuracy on ImageNet-1K under different values of** $L$. We replace the FFNs with MoE layers in the **last** $L$ ViT blocks. $L = 0$ represents the non-MoE DINOv2 baseline, and $L = 12$ indicates that every block contains the MoE layer.

accuracy consistently exhibits a trend of initially increasing and then decreasing, with this trend becoming more pronounced as $N$ increases. This phenomenon has also been mentioned in (Daxberger et al., 2023). We hypothesize that introducing multiple experts too early in the shallow ViT blocks leads to optimization difficulties, and the gating network struggles to achieve precise routing due to limited information (a more detailed analyze of this is given in Fig. 5). This suggests a potential *instability* in the design of ViMoE. Simply adding MoE layers to all ViT blocks without careful consideration may not lead to optimal results. A scan over different values of $L$ is required to determine the most suitable number of layers, which inevitably increases the design cost.

**Shared Expert for Stabilising ViMoE.** As previously discussed, the shared expert learns and consolidates knowledge from all the data, making it more effective in capturing common information. We consider this structure effective in alleviating challenges of gating decisions and the limitations of individual expert learning in sparse structures. Therefore, we attempt to incorporate the shared expert into ViMoE to mitigate the potential instability in training MoE layers. In Fig. 3 we present a comparison between models with and without shared expert. Incorporating the shared expert allows ViMoE to achieve stable results, eliminating the need for an exhaustive search to determine the optimal number of layers $L$. Even the naive approach of adding MoE layers to all ViT blocks yields good accuracy, preventing performance degradation caused by inappropriate MoE configurations. Additionally, with the inclusion of shared expert, ViMoE achieves a 0.4% improvement in accuracy (84.3% *vs.* 83.9%), and a **1.2%** increase compared to the DINOv2 baseline (83.1%).

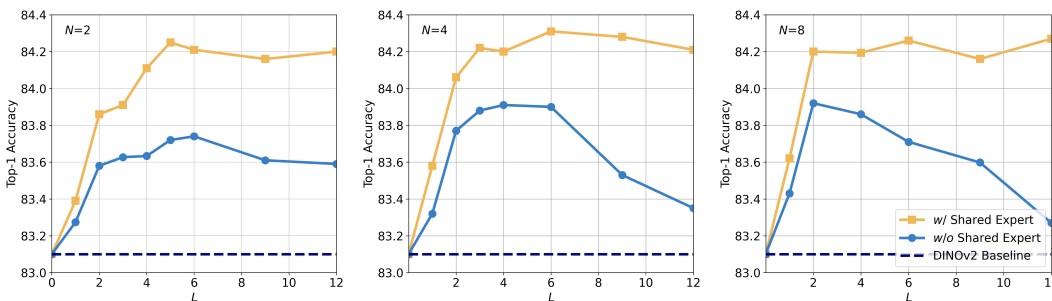

Figure 3: **Top-1 accuracy with and without the shared expert at different values of $L$.**

**Convergence Advantage.** Taking $N = 8$ and $L = 12$ as an example, Fig. 4 shows the training curves with and without shared expert, along with the DINOv2 baseline for reference. It is evident that simply adding sparse MoE layers slows down convergence in the early training epochs, and the final performance is nearly indistinguishable from the baseline, supporting the hypothesis that an improper MoE setting can even hinder optimization. In contrast, when shared expert is introduced, training becomes more stable, convergence is faster, and accuracy improves significantly. It is worth mentioning that, with the introduction of shared expert, the MoE layers contain a total of 9 experts (1 shared

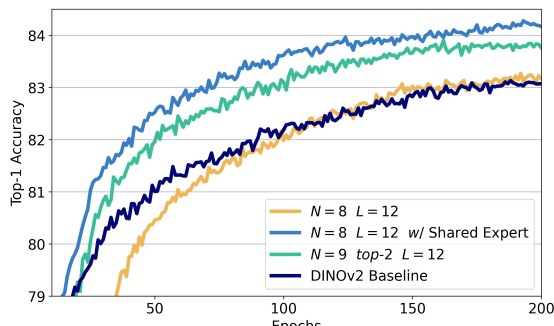

Figure 4: **Convergence curves** for training ViMoE under different configurations.

expert and 8 routed experts), and the forward pass activates both the shared expert and one selected routed expert. To ensure a fairer comparison, we conducted an ablation study by selecting the top-2 experts from the 9 routed experts. On one hand, selecting 2 out of 9 can be seen as a denser setup compared to selecting 1 out of 8, which partially mitigates the negative effects of being overly sparse. On the other hand, even with the same number of experts and activated experts, shared expert still demonstrates the advantage with faster convergence and higher accuracy.

### 3.2 Efficient Exploration Based on Stability

After constructing the stable ViMoE, we further analyze Fig. 3 and observe the presence of a performance plateau. Interestingly, the turning point differs for each $N$. For $N = 2$, $N = 4$, and $N = 8$, accuracy already surpasses 84.2% at $L = 5$, $L = 3$, and $L = 2$, respectively. Beyond these number of layers, no significant improvement is observed by adding more MoE. We attempt to explain these phenomena and propose strategies for designing a more efficient ViMoE.

**Routing Heatmap.** Taking $N = 8$ as an example, we plot the routing heatmaps of several MoE layers in Fig. 5. These heatmaps illustrate the distribution of class samples across different experts, helping us observe whether the experts are capable of capturing distinctive information. It can be observed that for the MoE layers in the shallow ViT blocks (*e.g.*, $l = 12$), the gating network struggles to consistently route images of the same class to the same expert or effectively distinguish the classes each expert should focus on. This indicates that the experts fail to learn highly discriminative knowledge; instead, they are likely performing similar functions, indiscriminately extracting common features. We then focus on the layer where the accuracy plateau occurs for $N = 8$, corresponding to $L = 2$. It is evident that in the last two MoE layers, the gating network can effectively assign the appropriate expert to each class, and the multiple experts can specialize in handling the corresponding data. Therefore, we conclude that the deep layers are where MoE truly achieves its *divide-and-conquer* objective, with different experts specializing in handling class-specific content. This observation validates the empirical approach of placing MoE layers in the last few ViT blocks (Wu et al., 2022; Liu et al., 2024) as a reasonable strategy. In contrast, MoE struggles to

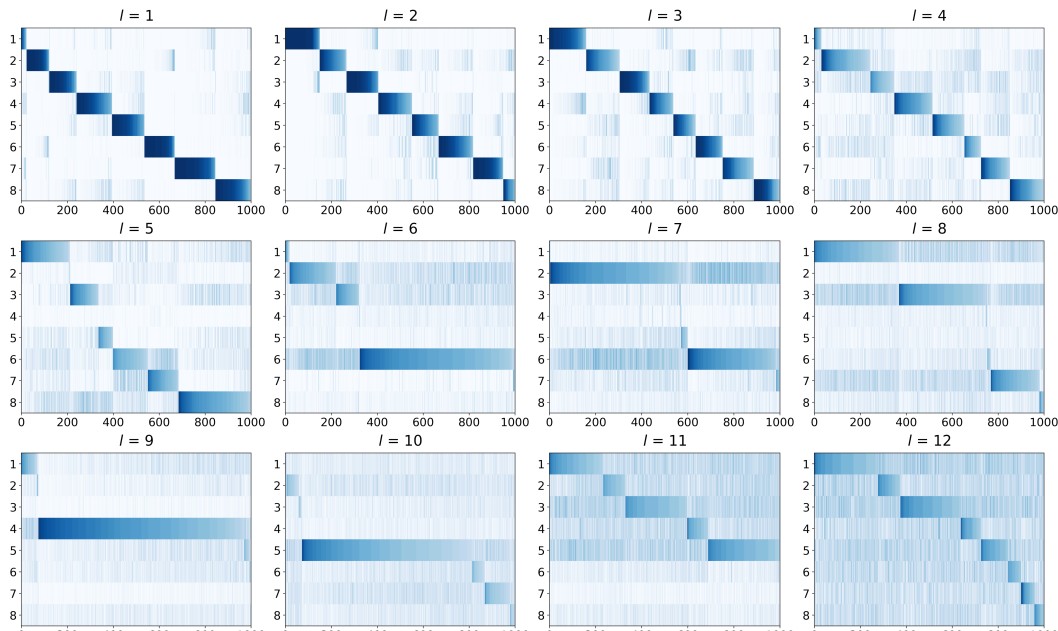

Figure 5: **Routing heatmap for the $l$-th MoE layer,** where $l = 1$ represents the deepest (last) layer and $l = 12$ denotes the shallowest (first) layer. The $x$-axis is the class ID from ImageNet-1K, and the $y$-axis is the expert ID. The label order in each figure is adjusted for better readability. Darker colors indicate a higher proportion of images from the corresponding class routed to the expert.

demonstrate its advantages in the shallow ViT blocks, as the use of multiple experts seems unnecessary for capturing basic visual features. The sparse structure may instead introduce optimization difficulties, making the original dense FFN structure a simpler and more suitable choice.

**Routing Degree.** Another interesting observation is that the number of MoE layers $L$ required varies with the number of experts $N$. We suggest this is related to the routing degree, which represents the number of possible expert combinations and can be simply defined as $D = (C_N^k)^L$. Since we fix the gating selection to top-1 (*i.e.*, $k = 1$), we obtain $D = (C_2^1)^5 = 32$ for $N = 2$, $D = (C_4^1)^3 = 64$ for $N = 4$, and $D = (C_8^1)^2 = 64$ for $N = 8$. This implies that approximately 32 to 64 routing combinations are sufficient for effectively partitioning and processing the data. Fewer combinations may affect performance, while more do not yield further significant gains. From another perspective, if we view the gating network allocating experts to data as a clustering process, the routing degree essentially reflects the number of clusters formed from the dataset. Each expert combination can then specialize in learning from the samples of its corresponding cluster, facilitating the model in reaching optimal effectiveness. Our results validate that end-to-end training can effectively achieve this clustering effect, without the need for additional clustering strategies to provide prior information for the gating mechanism (Liu et al., 2024).

**Efficient ViMoE.** The above conclusions are drawn from scanning the number of MoE layers. From another perspective, we can approximately predict the routing degree by observing the expert allocation in each layer. As illustrated in Fig. 5, the routing heatmap provides evidence of which MoE layers play a critical role, potentially indicating the necessary expert combinations that impact the results. These insights guide us in refining the structural design, retaining the essential MoE layers while removing the unnecessary ones, thereby developing a more efficient ViMoE. Moreover, we expect these findings are not limited to the ImageNet-1K dataset. In Sec. 4.2, we further explore the transfer of these insights to CIFAR100 (Wang et al., 2017) to validate their **generality**.

In Table 1, we present various ViMoE configurations and compare their parameter counts. Although sparse MoE layers increase the total number of parameters, since we set the gate to route each image to the top-1 expert, it achieves higher accuracy without increasing the activated parameter counts or the inference burden. With the inclusion of the shared expert, we further improve accuracy at relatively low extra cost. For example, when $N = 8$ and $L = 2$, only **2.4M** additional activated

Table 1: **Model efficiency.** The model sizes, inference burden, and ImageNet-1K accuracy of ViMoE. All models are based on ViT-S/14. $L = 0$ refers to the DINOv2 baseline. FLOPs metric is evaluated using $224 \times 224$ image resolution.

| $N$ | $L$ | w/ Shared Expert | Total Param. | Activate Param. | FLOPs | Acc. |
|---|---|---|---|---|---|---|
| - | 0 | - | 22.0M | 22.0M | 5.53G | 83.1 |
| 2 | 5 | | 27.9M | 22.0M | 5.53G | 83.6 |
| 2 | 5 | ✓ | 33.8M | 27.9M | 7.04G | 84.3 |
| 2 | 12 | ✓ | 50.4M | 36.2M | 9.17G | 84.2 |
| 4 | 3 | | 32.7M | 22.0M | 5.53G | 83.9 |
| 4 | 3 | ✓ | 36.2M | 25.6M | 6.44G | 84.2 |
| 4 | 12 | ✓ | 78.8M | 36.2M | 9.17G | 84.2 |
| 8 | 2 | | 38.6M | 22.0M | 5.53G | 83.9 |
| 8 | 2 | ✓ | 40.9M | 24.4M | 6.13G | 84.2 |
| 8 | 12 | ✓ | 135.5M | 36.2M | 9.17G | 84.3 |

Table 2: **Top-1 accuracy on ImageNet-1K.** All models are evaluated at resolutions $224 \times 224$. We select $N = 8$, $L = 2$ as a representative configuration to report. $\star$ indicates the inclusion of the shared expert.

| Method | Arch. | Activate Param. | FLOPs | Acc. |
|---|---|---|---|---|
| DINO | ViT-S/16 | 22.1M | 4.25G | 81.5 |
| BEiT | ViT-S/16 | 22.1M | 4.25G | 81.7 |
| iBOT | ViT-S/16 | 22.1M | 4.25G | 82.3 |
| DINOv2 | ViT-S/14 | 22.0M | 5.53G | 83.1 |
| DINO | ViT-B/16 | 86.6M | 17.58G | 82.8 |
| MoCov3 | ViT-B/16 | 86.6M | 17.58G | 83.2 |
| BEiT | ViT-B/16 | 86.6M | 17.58G | 83.4 |
| MAE | ViT-B/16 | 86.6M | 17.58G | 83.6 |
| iBOT | ViT-B/16 | 86.6M | 17.58G | 84.0 |
| ViMoE | ViT-S/14 | 22.0M | 5.53G | 83.9 |
| ViMoE$\star$ | ViT-S/14 | 24.4M | 6.13G | **84.2** |

Table 3: **Comparison between dense structure and sparse MoE.** For dense structures, $L$ indicates that each of the last $L$ layers contains two FFNs.

| Arch. | $L$ | $N$ | Activate Param. | FLOPs | Acc. |
|---|---|---|---|---|---|
| Dense | 0 | - | 22.0M | 5.53G | 83.1 |
| Dense | 2 | - | 24.4M | 6.13G | 83.6 |
| Dense | 3 | - | 25.6M | 6.44G | 83.8 |
| Dense | 5 | - | 27.9M | 7.04G | 83.8 |
| Dense | 12 | - | 36.2M | 9.17G | 83.9 |
| Sparse | 2 | 8 | 24.4M | 6.13G | 84.2 |
| Sparse | 3 | 4 | 25.6M | 6.44G | 84.2 |
| Sparse | 5 | 2 | 27.9M | 7.04G | 84.3 |

Table 4: **Ablation studies of different routing strategies.** We calculate the average number of routed experts and activated parameters per image, with the total number of experts being $(N + 1) \times L$ (including the shared expert).

| Strategy | $L$ | $N$ | Avg. # Experts | Activate Param. | Acc. |
|---|---|---|---|---|---|
| Token | 2 | 8 | 16.3 | 38.9M | 84.1 |
| Token | 3 | 4 | 14.4 | 35.5M | 84.2 |
| Token | 5 | 2 | 14.8 | 33.6M | 84.1 |
| Image | 2 | 8 | 4 | 24.4M | 84.2 |
| Image | 3 | 4 | 6 | 25.6M | 84.2 |
| Image | 5 | 2 | 10 | 27.9M | 84.3 |

parameters are required to surpass the baseline by **1.1%** in accuracy. Furthermore, a comparison with $L = 12$ highlights the efficiency of our structural design for ViMoE, significantly reducing parameter count without sacrificing accuracy.

# 4 EXPERIMENTS

## 4.1 IMAGE CLASSIFICATION ON IMAGENET-1K

**Implementation Details.** All experiments are conducted on the DINOv2 (Oquab et al., 2023) pretrained ViT-S/14 (Dosovitskiy, 2020) and fine-tuned on ImageNet-1K (Deng et al., 2009) with $224 \times 224$ image resolution for 200 epochs. By default, we use the AdamW (Sun et al., 2021) optimizer with a batch size of 1024, a weight decay of 0.05, and a layer-wise learning rate decay of 0.65. The peak learning rate is set to $1e^{-4}$ with a warm-up of 20 epochs. For the MoE layers, we configure three different numbers of experts ($N = 2$, $N = 4$, and $N = 8$), selecting the top-1 expert, with the load balancing loss coefficient $\alpha$ set to 0.01.

**Results.** Most of the empirical results on the ImageNet-1K benchmark have already been presented earlier. Here, we compare ViMoE against various self-supervised models (Bao et al., 2021; Zhang et al., 2022; Zhou et al., 2021; Oquab et al., 2023; Xinlei et al., 2021; He et al., 2022). As shown in Table 2, ViMoE achieves an 83.9% top-1 accuracy, which is **0.8%** higher than DINOv2 without increasing activated parameters. With shared expert, the accuracy further improves to 84.2%, outperforming DINOv2 by **1.1%**. Notably, we achieve this performance using only ViT-S/14, surpassing other methods based on ViT-B/16, while activating less than one-third of their parameters.

**Comparison with Dense Structures.** Previous results validate the advantage of the MoE structure over dense models. However, when we introduce the shared expert, the number of activated parameters increases. To ensure fairness, we attempt to modify the DINOv2 baseline by aligning the number of activated parameters while maintaining a dense architecture. One feasible approach is to mimic the MoE by setting two experts and selecting the top-2, which allows for the addition of an extra FFN in the ViT block.

In Table 3, we present the results of dense structure with different layer counts and compare them with sparse MoE. While increasing the number of parameters provides accuracy gains, the sparse structures are obviously more efficient and have a higher upper bound. For instance, at $L = 2$ with 24.4M activated parameters, sparse MoE outperforms the dense one by 0.6%.

**Routing Distribution.** In Sec. 2.1, we introduce the load balancing loss to assist in training sparse MoE models. Its purpose is to ensure that multiple experts receive inputs more evenly, preventing the majority of data from being routed to a single expert and thus avoiding the model from degrading into a dense structure. We calculate the proportion of data allocated to each expert in the MoE layers, as shown in Fig. 6. It is evident that the gating network distributes the data relatively evenly across multiple experts. Combined with the observations from Fig. 5, this validates the expectation that MoE layers enable different experts to handle specific information.

**Routing Strategy.** In Sec. 2.2, we introduce the routing strategy, where experts are selected for the entire image rather than for each token. In Table 4, we conduct an ablation study comparing these two strategies, showing no significant difference in accuracy. This indicates that the image-level strategy, while simpler, is effective because it aligns with the task objective of

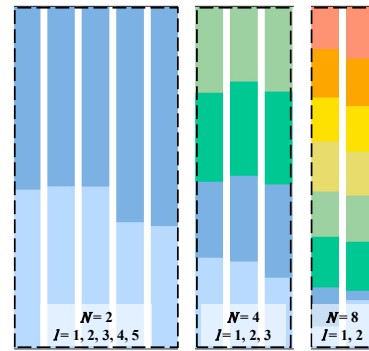

Figure 6: **Distribution of expert loadings.** Different colors represent different experts.

image classification. Additionally, we calculate the average number of routed experts and activated parameters per image, further confirming that our choice is more efficient.

## 4.2 Validation on CIFAR100

The above-mentioned observations and conclusions are based on ImageNet-1K (Deng et al., 2009). To demonstrate generalizability, we conduct validation on CIFAR100 (Wang et al., 2017) and aim to identify the most suitable ViMoE configuration.

**Implementation Details.** All models are fine-tuned on CIFAR100 for 100 epochs with a weight decay of 0.3. The peak learning rate is set to $3e^{-4}$ with a warm-up of 3 epochs, while all other settings remain consistent with those adopted on ImageNet-1K.

**Baseline and Stable ViMoE.** First, we use the DINOv2 (Oquab et al., 2023) self-supervised pre-trained ViT-S/14 (Dosovitskiy, 2020) and fine-tune it on CIFAR100 as the baseline, which achieves a top-1 accuracy of 91.3%. Next, we convert the ViT blocks into the ViMoE framework. Considering that CIFAR100 has fewer classes and samples than ImageNet-1K, we set the number of experts to $N = 4$ in our experiments. Based on prior experience, ViMoE with the shared expert tends to yield stable results, allowing us more flexibility in setting the number of MoE layers. We opt for the most straightforward approach by adding MoE layers to every block, *i.e.*, $L = 12$. Under this setting, ViMoE achieved a top-1 accuracy of **91.6%**, surpassing the baseline by 0.3%. Additionally, we compare the model without the shared expert, which yields an accuracy of only 78.4%, falling far short of the baseline. This demonstrates that MoE is not a simple design that guarantees stable gains. In fact, the optimization complexity introduced by sparse structures in certain ViT blocks may have significant negative impacts, further highlighting the necessity of designing a stable ViMoE.

**Efficient Structures Derived from Observations.** We observe the behavior of MoE within the stable ViMoE and further analyze which layers play a critical role. Following the approach outlined in Sec. 3.2, we generate the routing heatmaps, as shown in Fig. 7. It is evident that in the last two layers, *i.e.*, $l = 1$ and $l = 2$, the gating network clusters data classes effectively, allowing each expert to specialize in handling specific classes. In contrast, the shallower layers do not exhibit

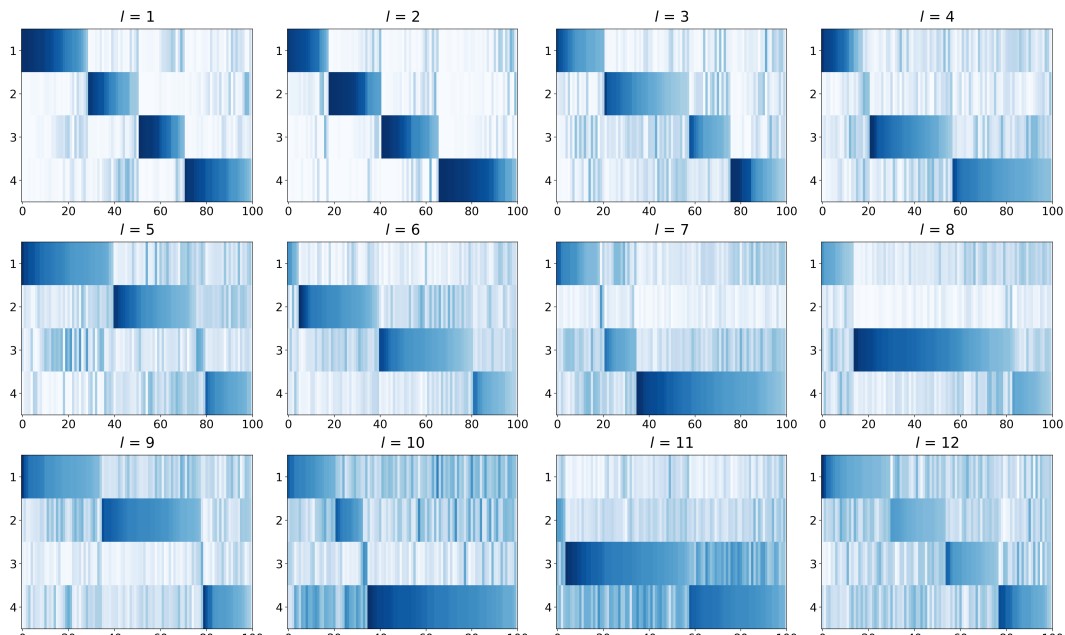

Figure 7: **Routing heatmap for the $l$-th MoE layer on CIFAR100.** The $x$-axis is the class ID, and the $y$-axis is the expert ID. The label order in each figure is adjusted for better readability. Darker colors indicate a higher proportion of images from the corresponding class routed to the expert.

Table 5: **Top-1 accuracy on CIFAR100 under different configurations.**

|  | $L=1$ | $L=2$ | $L=4$ | $L=6$ | $L=9$ | $L=12$ |
|---|---|---|---|---|---|---|
| *w/o Shared Expert* | | | | | | |
| $N=2$ | 91.4 | 91.5 | 91.5 | 91.5 | 91.3 | 91.2 |
| $N=4$ | 91.4 | 91.5 | 91.3 | 90.7 | 89.2 | 78.4 |
| $N=8$ | 91.5 | 91.3 | 90.8 | 89.9 | 80.9 | 52.9 |
| *w/ Shared Expert* | | | | | | |
| $N=2$ | 91.5 | 91.6 | 91.7 | 91.7 | 91.6 | 91.6 |
| $N=4$ | 91.6 | 91.7 | 91.7 | 91.7 | 91.7 | 91.6 |
| $N=8$ | 91.6 | 91.6 | 91.7 | 91.7 | 91.7 | 91.5 |

clear expert specialization, suggesting that these MoE layers may not be necessary and that a single FFN can replace the role of multiple sparse experts. Based on this, we estimate the routing degree for CIFAR100 to be around 4 to 16. To validate this hypothesis, we experiment with the $L=2$ configuration, achieving an accuracy of **91.7%**. This setup maintains good results while reducing parameters and improving efficiency.

**Layer Scanning.** We further validate the results by layer scanning, as shown in Table 5. When no shared experts are employed, an unreasonable configuration of the number of MoE layers leads to significantly lower accuracy, which is even more pronounced than what we observed in ImageNet-1K. We attribute this to the fact that on datasets with smaller data volumes and fewer classes, overly sparse architectures hinder each expert from being sufficiently optimized. These results reinforce the necessity of incorporating shared experts to stabilize model convergence. Moreover, for the efficient ViMoE, the required routing degree (*i.e.*, the number of expert combinations) is indeed smaller when the dataset contains fewer classes. It can be observed that incorporating MoE only in the deepest one or two layers is sufficient to achieve considerable accuracy.

**Discussion.** Comparing the CIFAR100 results with those from ImageNet-1K, we observe that fewer experts are required when there are fewer classes. This aligns with the intuition that having numerous experts handle simpler tasks does not provide additional benefits and may even introduce drawbacks. Therefore, training a smaller number of experts to be specialized and efficient is sufficient.

## 5 RELATED WORK

**Mixture-of-Experts (MoE)** model is first introduced in (Jacobs et al., 1991) and has been widely studied for its ability to modularize learning and reduce interference across data domains (Zhou et al., 2022; Rajbhandari et al., 2022). MoE uses a gating network to assign which experts should handle each data sample. Early MoE models were densely activated, meaning every input triggered all experts, which, while functional, was computationally expensive due to the significant resources required to process each input through all experts (Masoudnia & Ebrahimpour, 2014). Modern mainstream MoE models can be regarded as an application of dynamic neural networks (Han et al., 2021), using sparse activation selecting only a subset of experts to handle each input, which greatly reduces computational costs while preserving model expressiveness and performance (Hwang et al., 2023; Hazimeh et al., 2021). This approach has become increasingly important in large language models, where efficiency and scalability are paramount. Notable works in NLP, such as Switch Transformers (Fedus et al., 2022), GShard (Lepikhin et al., 2020), and GLaM (Du et al., 2022), have successfully applied sparse MoE, demonstrating significant advancements in handling large-scale tasks while optimizing resource usage.

**MoE for Vision Tasks.** In recent years, the high efficiency of MoE in NLP tasks has motivated researchers to explore their applications in the visual domain. Works such as V-MoE (Riquelme et al., 2021) and $M^3$vit (Fan et al., 2022) integrate sparse MoE architectures into Vision Transformers. By replacing certain dense feedforward layers with sparse MoE layers, these models achieve efficient modeling in image classification tasks, enhancing computational efficiency and performance. Simultaneously, pMoE (Chowdhury et al., 2023) and DiT-MoE (Fei et al., 2024) introduce sparse conditional computation mechanisms. Specifically, pMoE employs CNNs as experts, dynamically selecting image patches for each expert, thereby reducing computational costs while maintaining generalization performance. DiT-MoE optimizes input-dependent sparsity in large diffusion transformer models, improving the efficiency and performance of image generation. Additionally, AdaMV-MoE (Chen et al., 2023) and the work by (Wu et al., 2022) focus on multi-task visual recognition and efficient training of large MoE vision transformers.

**Transformer for Vision.** Transformer models initially achieved remarkable success in natural language processing and was later introduced into computer vision, leading to the development of Vision Transformers (ViT). Vision Transformers (ViT) (Dosovitskiy, 2020) introduced a new approach to image processing by dividing images into patches and treating them like words in text, allowing for global feature extraction across the entire image. Unlike convolutional neural networks (CNNs) that rely on local receptive fields, ViT's Transformer-based architecture captures broader context, achieving performance on par with, or exceeding, that of CNNs. In the realm of self-supervised learning, MoCov3 (Xinlei et al., 2021) extended the momentum contrastive learning approach to ViT, successfully training high-quality visual features from unlabeled data. Inspired by BERT's (Kenton & Toutanova, 2019) masked language modeling, methods such as BEiT (Bao et al., 2021), MAE (He et al., 2022), and iBOT (Zhou et al., 2021) pre-train ViTs through masked image modeling to enhance the model's generalization ability and representation learning. DINOv2 (Oquab et al., 2023) employed self-supervised learning methods based on knowledge distillation, utilizing larger datasets and longer training periods, allowing it to learn robust visual features in an unsupervised manner, further advancing self-supervised ViT.

## 6 CONCLUSION

In this work, we integrate the sparse Mixture-of-Experts (MoE) architecture into the classic Vision Transformer (ViT), termed ViMoE, to explore its potential application in image classification. We report the challenges encountered in designing ViMoE, particularly in determining the configuration of MoE layers without prior guidance, as inappropriate expert arrangements can negatively impact convergence. To mitigate this, we introduce the shared expert to stabilize the training process, thus streamlining the design by eliminating the need for repeated trials to find the optimal configuration. Furthermore, by observing the routing behavior and the distribution of samples across experts, we identify the MoE layers that are crucial for the divide-and-conquer processing of data. These insights allow us to refine the ViMoE architecture, achieving both efficiency and competitive performance. We hope this work provides new insights into the design of MoE models for vision tasks and offers valuable empirical guidance for future research.

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
