# OpenReview forum: "ViMoE: An Empirical Study of Designing Vision Mixture-of-Experts"
_ICLR.cc/2025/Conference — ICLR 2025 Conference Withdrawn Submission_

### Official Review · Reviewer_EbgX · 2024-10-23

**Soundness:** 2
**Presentation:** 2
**Contribution:** 1
**Rating:** 3
**Confidence:** 4

**Summary:**

The authors propose to study MoE configurations in ViTs, primarily for the task of image classification on ImageNET1k. They study the impact of using a varying number of MoE layers (rather than dense layers), and conclude that performance drops when using MoEs at too many early layers (when using a per-image expert assignment shared across all tokens). The paper then makes some suggestions about more efficient MoE design based on these findings. Furthermore, the paper also validates previous works’ ideas of using a shared expert in addition to image-specific experts.

**Strengths:**

- The study of optimal MoE configuration is an important issue. More and more models are turning to MoEs to increase either network capacity or for specialized subcomputations, and the authors’ object of study has potential to influence a lot of future model design in significant ways.
- Independent of my reservations outlined below in the weaknesses section, the experimental section is relatively thorough, and I appreciate the multiple ways in which the authors attempt to visualize and evaluate the expert loadings and assignments. In particular, I feel ImageNET1k is a smart choice made by the authors for MoE study, through being a large scale diverse dataset that nonetheless has ground-truth labels allowing one to quantify expert routing choices and thus study MoE design in a more systematic manner.

**Weaknesses:**

## [W1] Per-image expert assignment confounds conclusions

The major results in the paper are derived from experiments exploring the impact of varying which layers use MoEs for ImageNET classification in ViTs.

My main concern about the paper is that the authors’ design choice to share the expert assignment across *all* tokens in an image is problematic, and complicates their conclusions. I suspect this is underestimating the performance of networks trained with MoEs at earlier layers with a more sensible configuration (not tying expert assignments per image). Crucialy, a lot of conclusions in the paper seem to hinge on this result.

The authors state on [L080] that `in the shallow blocks, the gating network struggles to consistently route images of the same class to the same expert`. But this is the expected outcome--early in the network, the representations are not linearly separable by class (hence why we use more layers in a network, to transform representations to the point at which they can be linearly classified). However, the expert assignment itself is a linear transformation of the shallow features (plus softmax just for selection). Thus, we cannot hope to read off class information from such early layer representations, and forcing image-level expert assignments here is in conflict with the scale of the features extracted at shallow layers.

In contrast, existing work reports useful MoE expert specialization to lower-level patterns and textures shared across many images of different classes, when *not* making image-level assignments (see Fig. 4 of [1] and "expert 17" of Fig. 2 in [2]). This is consistent with the intuitive notion of a coarse- to fine-grained processing of information as the network deepens. Furthermore, as the authors mention on [L057], their design is a departure from the common approach of using a more flexible per-*token* assignment.

Whilst the authors do have a small study in Tab. 4 to explore these two strategies, I believe this must be controlled for through a larger ablation study in the main experiments for (e.g.) Fig. 2. If the authors remove the image-level assignment in Fig. 2's experiments, and yet still observe decreased performance with earlier MoE layers, this would be a useful finding of the paper. At the minute, however, I believe that the presence of this confounding design choice means we cannot validate the conclusions in the authors’ study, given that the possibility the bad performance is explained simply by the tying of the image-level expert assignments.

## [W2] Unclear contributions

In light of the questionable conclusions about the design choices made by the paper, I am unclear what the paper’s contributions are. For example, there are some interesting experiments on pruning MoE networks for efficiency, but these conclusions hinge upon the previous analysis of which layers benefit from MoEs and the assumption that earlier MoEs must make class-discriminative routings.

In addition to the study about layer placement, the authors do state in the conclusion that `To mitigate this [instability], we introduce the shared expert to stabilize the training process, thus streamlining the design by eliminating the need for repeated trials to find the optimal configuration.`. However, the “shared expert” is not a new feature proposed by the authors--the results in Figs 3&4 seem to simply confirm that previous works’ ideas of shared experts are beneficial, but we already know this from the original papers. Thus, I am struggling to see much value or new insights provided by the paper--I suggest the authors should clarify the novelty of their contributions, beyond confirming the benefits of the existing idea of using shared experts. For example, are there any analyses or insights here in the pruning study that are unique to this paper?

## [W3] Missing comparisons and discussions regarding previous work

The authors do not compare to nor discuss the recent important work of “From Sparse to Soft Mixtures of Experts” [3], which proposes improvements to MoE design specifically in the case of ViTs, exploring ImageNET1k classification like the authors.

How do the authors’ proposed design modifications (following their analysis) relate to the work in [3]? Can the two be combined for further improvements, or is one strictly dominant? A comparison to the two design choices would be necessary here to evaluate the authors’ MoE design contributions, as I understand [3] to be the current published SOTA approach for ViTs specifically.

---

**Minor**
* [L044] subsets -> subset?

---

[1]: Oldfield et al. “Multilinear Mixture of Experts: Scalable Expert Specialization through Factorization.” NeurIPS 2024.

[2]: Mustafa et al. “Multimodal Contrastive Learning with LIMoE: the Language-Image Mixture of Experts.” NeurIPS 2022.

[3] Puigcerver et al. “From Sparse to Soft Mixtures of Experts.” ICLR 2024.

**Questions:**

I have no major additional questions -- my primary concern is the major [W1] above, and I am happy to hear what the authors have to say about this.

---

### Official Review · Reviewer_RvqA · 2024-10-31

**Soundness:** 2
**Presentation:** 3
**Contribution:** 2
**Rating:** 3
**Confidence:** 5

**Summary:**

This paper conduct a comprehensive analysis of MoE's potential in vision tasks, specifically image classification. However, they find that the model's performance is highly sensitive to the configuration of MoE layers, posing challenges for achieving optimal results without careful layer design. The root issue lies in suboptimal MoE layers causing unstable routing, preventing experts from effectively capturing valuable knowledge. To address this, they introduce a shared expert designed to learn and retain common information, stabilizing ViMoE’s performance. Additionally, they present a method for analyzing expert routing behavior, identifying which MoE layers specialize in processing specific types of information.

**Strengths:**

1. The paper writing is clear and easy to understand.
2. They propose a new MoE for vision task, which is easy to implement.
3. The analysis is sufficient.

**Weaknesses:**

1. They only explore the image classification task.
2. The dataset they used is common and corrupted. For example, ImageNet 1k has lot of samples with bad annotations. You should also try some better and hard datasets.
3.  The network they proposed is not novel, which can be seen in many NLP papers.

**Questions:**

1. Could you test your methods in other datasets, such as SUN, Places?
2. Could you explain again about your novelty and the difference between your design and others?

---

### Official Review · Reviewer_dNFx · 2024-11-04

**Soundness:** 2
**Presentation:** 2
**Contribution:** 2
**Rating:** 3
**Confidence:** 4

**Summary:**

This paper explores fine-tuning DINOv2 for image classification tasks by replacing FFNs with MoE layers. It considers different numbers of MoE layers, and adding them to different ViT blocks. It also considers different MoE structures. Experiments show that fine-tuning DINOv2 with this method improves over DINOv2, and also surpasses other backbones with better efficiency.

**Strengths:**

- The shows that a simple method (adding sparse MoE layers, then fine-tuning) can improve the classification performance of pre-trained ViTs. It also presents a series of ablation study on various design choices, which could be helpful for future works.
- The main experimental results suggest that the proposed method is overall competitive.
- The presentation is clear and easy to understand.

**Weaknesses:**

- Unfair comparison is a serious issue of this paper. 1) The method itself is a plug-and-play module and should not be stated as a new method in Fig.1 and Tab.2. Instead, it should be marked as DINOv2+ViMoE to avoid confusion. 2) It should be ensured that all other reported methods have gone through 200-epochs fine-tuning on ImageNet, and results of ViMoE+X (other backbones) should be reported. 3) ViMoE should be compared with other fine-tuning (adaptation) techniques, eg, SupCon, VPT, LoRA, etc.
- Adding sparse MoE to ViTs during fine-tuning is of limited technical novelty. Despite the paper is stated to be an empirical study, the overall contribution is rather limited.
- Evaluation is limited to ImageNet, and it should be extended to more datasets to show generalization.
- Some technical details are missing in the text, eg, from L160 more details about the shared expert should be provided.

**Questions:**

Minor:

L414: $3e^{-4}$, should be $3\times10^{-4}$ or $0.0003$ or 3e-4.

---

### Official Review · Reviewer_gzs3 · 2024-11-04

**Soundness:** 2
**Presentation:** 2
**Contribution:** 1
**Rating:** 3
**Confidence:** 3

**Summary:**

The paper integrates the MoE structure into the Vision Transformer (ViT) to create ViMoE and studies its application in image classification. MoE models are promising for increasing capacity and scalability. However, the performance of ViMoE is sensitive to MoE layer configuration, as inappropriate layers cause unreliable routing and prevent experts from getting useful knowledge. To solve this, a shared expert is introduced to learn common information for stable ViMoE construction. Additionally, the analysis of expert routing behavior shows which MoE layers handle specific information and which don't, guiding the removal of redundancies to make ViMoE more efficient without losing accuracy. The work aims to provide new insights and empirical guidance for vision MoE model design in future research.

**Strengths:**

This article studies the Vision MoE architecture. The article is easy to read and has a clear structure.

**Weaknesses:**

My doubts regarding this paper primarily stem from two major aspects, namely the experimental setup and the theoretical foundation.

1. At present, the most prominent application domain of the MoE lies in Vision - Language Models (VLM) and Large - Language Models (LLM). In this paper, although the focus is on investigating the role of MoE in visual representation learning, there is a significant shortcoming as it fails to provide validation within practical application scenarios. Practical scenarios are crucial as they can accurately reflect how the proposed model would perform in real-world situations where various factors come into play. Without such validation, the practical value of the research remains uncertain.

2. Moreover, the scale of the experiments conducted in this paper is far too limited. It predominantly relies on the CIFAR dataset. This narrow choice of dataset restricts the generalizability of the results. In the field of computer vision research, a more diverse range of datasets, including those with complex and large-scale images, should be used to comprehensively evaluate the performance of the model. Experiments based solely on CIFAR can hardly provide a comprehensive understanding of how the model would handle different types of visual data, thereby rendering the experimental conclusions rather insignificant.

3. Another major flaw is the absence of necessary comparisons regarding feature clustering visualization. Visualization of feature clustering can provide valuable insights into how the model organizes and processes visual information. It allows researchers to observe the patterns and structures within the data that the model has captured. Without such visual comparisons, it becomes difficult to assess the effectiveness of the proposed model in handling different visual features and understanding how it differentiates between various classes of objects within the images.

4. The performance improvement achieved by the scheme proposed in this paper is extremely marginal. A significant performance improvement is one of the key indicators of the value of a research contribution. When the enhancement in performance is so small, it raises questions about the practical utility and innovation of the proposed approach. It implies that the proposed model may not bring substantial benefits compared to existing methods in the field.

5. From a theoretical perspective, this paper lacks any form of in-depth theoretical or intuitive analysis. A solid theoretical basis is essential as it provides the rationale behind the proposed model and its design decisions. Intuitive analysis, on the other hand, helps in making the model more understandable and interpretable. Without these, the paper fails to establish a strong foundation for the proposed approach, leaving readers with doubts about the validity and soundness of the research.

To sum up, considering all these issues, I firmly believe that the quality of this paper does not meet the high standards required for ICLR.

**Questions:**

See the weakness part.

---

### Author Response · Authors · 2024-11-14

We sincerely appreciate the reviewers for dedicating their time and effort to reviewing our work and for recognizing the potential contributions we may have made to the design of vision MoE models. The insightful feedback, constructive comments, and suggestions provided by the reviewers have significantly enhanced the quality and clarity of our work. We will incorporate these valuable suggestions into the revised version to further improve the content.

---

### Note · Authors · 2024-11-14

I have read and agree with the venue's withdrawal policy on behalf of myself and my co-authors.